# Maternal complication of instrumental vaginal delivery and its associated factors: Systematic review and meta-analysis

**Hinsermu Bayu Abdi**[1]*, **Elias Bekele Wakwaya**[1], **Beyene Sisay Damtew**[1], **Beker Ahmed Hussen**[1], **Teresa Kisi Beyen**[2]

1 Department of Midwifery, College of Health Sciences, Arsi University, Assela, Ethiopia, 2 School of Public Health, College of Health Sciences, University of Gondar, Gondar, Ethiopia

☉ These co-authors contributed equally to this work.
* henybayu1@gmail.com

## Abstract

### Background

While instrumental vaginal delivery is generally a safe procedure, but it is associated with significant risks for both the mother and the newborn in developing countries. However, burden of maternal instrumental delivery complications and its predictors in Ethiopia is highly varied and there isn't data that indicate nation-level cumulative evidence. Therefore, this study aimed to assess pooled prevalence of maternal complication related to instrumental vaginal delivery and its associated factors among mothers who underwent instrumental vaginal delivery in Ethiopia.

### Methods

In this study, we conducted a search on PubMed, Scopus, Cochrane library, HINARI, and Google Scholar academic databases for studies published until August 2024. Keywords such as instrumental delivery, forceps, vacuum, complication, factors and Ethiopia were used to access literatures from the databases. For quality assessment and data extraction, The Joanna Briggs Critical Appraisal Tools and the Preferred Reporting Items for Systematic Reviews and Meta-Analyses (PRISMA) guidelines were utilized. Random-effect model was used to calculate the pooled prevalence of maternal complication of instrumental vaginal delivery. Subgroup analyses were also conducted to explore potential heterogeneity. The publication bias was assessed using Funnel plot and Egger's.

### Results

A total of 12 studies with 3745 study participants were participated in the present meta-analysis. The pooled prevalence of maternal complication of instrumental vaginal delivery was 21% (95% CI, 15.0%-28.0%). The prevalence of maternal complication was significantly different among studies conducted between ([2015–2020], and those conducted between [2020 and 2024]. Lowest (14%) in studies conducted before 2020 and highest (26%) among studies conducted 2020 and after (P-value<0.001). Type

**Data availability statement:** All relevant data are within the paper and its Supporting Information files.

**Funding:** The author(s) received no specific funding for this work.

**Competing interests:** The authors have declared that no competing intersts exist.

of instrument (AOR: 1.99, 95% CI: 1.37–2.90), episiotomy status (AOR: 3.49, 95% CI: 2.12–5.76), birthweight (AOR: 3.06, 95% CI: 1.88–4.97) and parity (AOR = 2.96, 95%CI: 1.80–4.85) were the factors associated with maternal complication of instrumental vaginal delivery.

## Conclusion

Our study shows that approximately one in five mothers who underwent instrumental vaginal delivery develop serious maternal complication. Type of the instrument, episiotomy status, birthweight and parity were important predictors of the maternal complication of instrumental vaginal delivery. Effective evaluation of indication, contra indication and pre-condition for each instrument helps to prevent the maternal complication of instrumental vaginal delivery.

## Trial registration

Registered in PROSPERO with ID: CRD42022366360.

## Background

An instrumental vaginal delivery (IVD) is a type of obstetric procedure with a purpose to deliver a baby using a forceps or a vacuum extractor to provide direct traction on the fetal skull [1–3]. Globally, about 10–20% of childbirths require intervention during delivery, and 6–12% of these interventions involve IVD [4,5]. In Ethiopia, 10% of all deliveries are conducted using either forceps or vacuum extractors [6].

The procedure is indicated based on maternal health status, labor progress and/or fetal conditions. Maternal indications for IVD include a prolonged labor, preeclampsia with severity sign, eclampsia, poor labor progress due to maternal exhaustion or fatigue, and an elective second stage labor shortening. Fetal indications include abnormal fetal heart rates or non-reassuring fetal status during the second stage of labor [6,7].

Although IVD is generally a safe procedure, it is associated with significant risks for both the mother and the baby in developing countries. In Ethiopia, the rate of IVD complications varies widely (4%–43%), and no data currently indicate the national-level or cumulative impact of IVD on maternal outcomes [3,8].

A significant number of mothers and their newborns experience a varying degree of avoidable morbidity and even mortality due to IVD complication. Laceration of the vagina, episiotomy extension, perineal tears, cervical laceration, traumatic postpartum hemorrhage, damage to the anal sphincter, uterine rupture, and pelvic floor damage are examples of maternal complications related to IVD [3,6,8].

There are inconsistencies among studies regarding the predictors of maternal complication related to IVD in Ethiopia. Type of instrument reported as a significant predictor of maternal complication related to IVD [3,8,9], However, other studies didn't report the association between this variable and maternal complication related to IVD [10,11].

Two independent studies conducted by Zenebe et al. and Eskinder et al. reported big baby (≥4.0 kg) as a strong predictor of maternal complication related to IVD [12,13], however birth weight of the newborn is not significantly associated with maternal complication related to IVD in vast majority of literatures [2,14,15]. There is literature that concludes: operator's experience and knowledge matter more than the instrument itself. However, the vast majority of published studies either omitted this factor or found it insignificant [8,10–12]. It is obvious

that a comprehensive study integrating prior research is needed to guide the decision-making process. Therefore, this systematic review and meta-analysis was carried out to provide a reliable pooled prevalence of maternal complications of IVD and its predictors at national level in Ethiopia.

## Materials and methods

### Study protocol and registration

The 2020 PRISMA Checklist was used to report the review findings [16]. The review was registered in the International Prospective Register of Systematic Reviews (PROSPERO) under the ID CRD42022366360.

### Search strategy

The studies were accessed through an electronic web-based search strategy from PubMed, SCOPUS, HINARI, the Cochrane Library for Systematic Reviews, Google search engine and WHO websites. Additionally, the Arsi University College of Health Sciences Library was searched for potential unpublished literature to reduce small research effect (publication bias). In conducting this review, a three-step search approach was used. A preliminary search of PubMed and Google scholar was conducted followed by an analysis of the text words in the title and abstract, as well as the index terms used to describe the article. All selected keywords and index terms were then used in a second search across all included databases.

The search strategy included a combination of keywords and Medical Subject Headings (MeSH) for the concepts of complication, Instrumental Delivery and Predictors. Accordingly, the following search strategy was used in PubMed and then adapted to other databases: (((((((((((((("Obstetric Labor Complications"[Mesh]) OR (Complication*[tw])) OR ("Maternal Complication*"[tw])) OR ("fetomaternal complication"[tiab:~0])) OR (Outcome[tw])) OR ("unfavorable birth outcomes"[tw])) OR ("feto-maternal outcome"[tw])) OR ("neonatal effects"[tw])) OR ("neonatal outcome*"[tw])) OR ("Fetal complications"[tw])) AND ((((((((("instrumental vaginal deliveries"[tw]) OR ("instrumental vaginal delivery"[tw])) OR ("Operative Vaginal Delivery"[tw])) OR ("Operative Vaginal Deliveries"[tw])) OR (Forceps-[tw])) OR ("vacuum delivery"[tw])) OR ("vacuum-assisted deliveries"[tw])) OR ("operative deliveries"[tw]))) AND (("Risk Factors"[Mesh]) OR ("associated factors" [tw])) OR (Determinants[tw] OR (predictor*[tw]))) AND (((((Ethiopia) OR ("Developing Country")) OR ("Low Income Country")) OR (East Africa)).

Search terms were based on PICO principles to retrieve relevant articles through the databases mentioned above. The search period was up to August 2024.

Thirdly, only studies in English-language were included in the reference list searches of all recognized papers and articles. Manual search of local libraries was conducted to review the topic.

### Study selection

Initially, duplicate articles were identified and removed from the records collected from the selected repositories, which were then imported into the EndNote X8 library. Three reviewers (HB, BS, and EB) independently screened the titles and abstracts of the articles in the shared EndNote library. Discrepancies among reviewers were resolved through discussion or by involving a third reviewer (EB). Subsequently, a full-text review was conducted independently by four reviewers (HB, BS, EB, and BA). The study selection process was summarized using a PRISMA flow diagram.

### Inclusion/exclusion criteria

The inclusion criteria included: (1) Cohort, case control and cross-sectional studies, (2) studies with effect estimates of maternal complication of IVD and Predictors, (3) written in English and (4) conducted in Ethiopia.

The exclusion criteria were: (1) studies conducted in non-human participants, (2) duplicate studies, (3) unavailable full text, and (4) commentaries and reviews.

### Outcome

The pooled prevalence of maternal complications related to IVD was the primary outcome of interest. The maternal complication was defined as a presence of at least one of the following adverse outcomes following IVD: perineal tear of cervix, laceration of cervix, laceration of vagina, extension of episiotomy, traumatic primary PPH, urinary retention, or uterine rupture) [3,12]. Additionally, the odds ratio of predictors, along with their 95% confidence intervals (CI) were extracted from the original studies to compute the pooled odds ratio of predictors.

### Data extraction

Data were extracted by two independent reviewers (HB and BA) using Joanna Briggs Institute Meta-Analysis of Statistics Assessment and Review Instruments in Microsoft excels 2010 format [16]. The extracted data included specific details such as study authors, year of publication, study region, study setting, study design, sample size, study respondents and predictor variables.

### Quality appraisal

To evaluate the study's methodological quality and determine how effectively it addressed potential biases in its design, quality appraisal was conducted independently by three reviewers (BA, EBand TK) using the JBI critical appraisal tool for cross-sectional studies. JBI was selected to evaluate the study's methodological quality and determine how effectively it addressed potential biases in its design [16]. Discrepancies were resolved through discussion led by the third author (BA). An 8-parameter critical appraisal tool with a possible response of "Yes", "No", "Unclear" and "Not applicable" was used for prevalence studies. The total score of the study was determined by summing up the "Yes" responses, yielding a maximum score of 8 points.

### Statistical analysis

Data entry was done using Microsoft Excel 2013 and subsequently imported into R software (version 4.1.3) for further analysis using the meta -package. The potential heterogeneity among the studies was assessed using the $I^2$ index. Due to the assumed high heterogeneity ($I^2 \geq 75\%$) between the studies, a random-effects model was employed for the meta-analysis. For the fixed effect model the Mantel Haenszel (MH) method was used, while the inverse variance method was applied for the random-effects model to pool the summary measure (odds ratio or Proportion). The between-study variance ($\tau^2$) was estimated using the Der Simonian and Laird method for the random effect model [17].

Subgroup analyses was conducted based on the year of publication (2015–2020 and 2020 to August 2024), geographical regions (Oromia, Amhara, T/Addis Ababa, and SNNRP), study setting (Tertiary hospital, General hospital, mixed facility) and the study quality. Multivariate meta-regression analysis was conducted using sample size and year of publication to explore

sources of heterogeneity in the pooled estimates. To assess the impact of an individual study on the pooled estimate of the effect size, a heterogeneity-based sub group sensitivity analysis was carried out [17]. Funnel plot was drawn to assess publication bias. Additionally, the egger test with p-value of <0.05 was used as evidence for significant publication bias among the selected studies. Trim and fill strategy was also conducted to input unpublished studies with less significancy. Odds ratio (OR) was used to describe the possible association between the outcome variable and its predictors. R and R studio software was used to conduct all analyses. The results were presented in a forest plot as a point estimate with 95% confidence intervals. Data were presented by developing clear descriptive summary using graphs, diagrams or tabular forms. Details like name of author, year of publication, study design, and number of participants were used to construct the summary table.

### Data quality control measures

The review authors sought to reduce risk of publication bias by conducting a search beyond published literature. The Joanna Briggs Institute Meta-Analysis of Statistics Assessment and Review Instruments was used to objectively evaluate the quality of included studies. Quality appraisal was conducted by senior researchers and the minimum quality score is 6 out of 8 (75%) indicating that they are low risk and included in the analysis.

## Results

### Study selection

A total of 9984 articles were identified from various data bases using our search strategies. Of these, 4113 articles were automatically removed due to duplication. Among the remaining 5871 studies, 5763 were excluded after a review of the title and abstract, leaving 108 studies were assessed for full-text review. Finally, only 12 articles fulfilled the eligibility criteria were included (Fig 1).

### Studies characteristics

The twelve included studies were published from 2015 to 2024, with sample sizes ranging from 209 to 406. All of the studies employed a cross-sectional study design. Regarding the study regions, 4 studies were conducted in Oromia [8,10,11,14], 3 in South nation and nationality people [2,15,18], 2 in Amhara [3,12], 1 in Tigray [19],1 in Sidama [20] and 1 study in Addis Ababa, the capital of Ethiopia [13]. More specifically, 4 studies conducted at tertiary hospitals [8,10,12,15], 4 conducted at general hospitals [11,18–20] and 4 conducted at mixed facilities ranging from health center to tertiary hospitals [3,8,13,14].

Parity was reported for only 2692 (71.9%) of respondents. Among mothers who underwent IVD, 1427 (38.1%) of them were primiparous. A majority, 2017 (53.9%) of the participants in the original studies, underwent delivery using forceps, while the remaining 1728 (46.3%) were attended with vacuum extractor. The twelve studies were assessed for quality, with the lowest quality score being 6 out of 8 on the quality scale, equivalent to 75.0%. This indicates a low risk of bias, and all studies were included in the analysis (Table 1).

### Pooled prevalence of maternal complication related to IVD

The 12 studies which involved 3745 participants, have estimated the pooled prevalence of maternal complication related to IVD to be 21% [95% CI, 15–28, $I^2$ = 97%, P < 0.01]. The lowest prevalence of maternal complication of IVD (4%) was reported by Hubena Z et al., while the highest prevalence (43%) was reported by Israel E et al. (Fig 2).

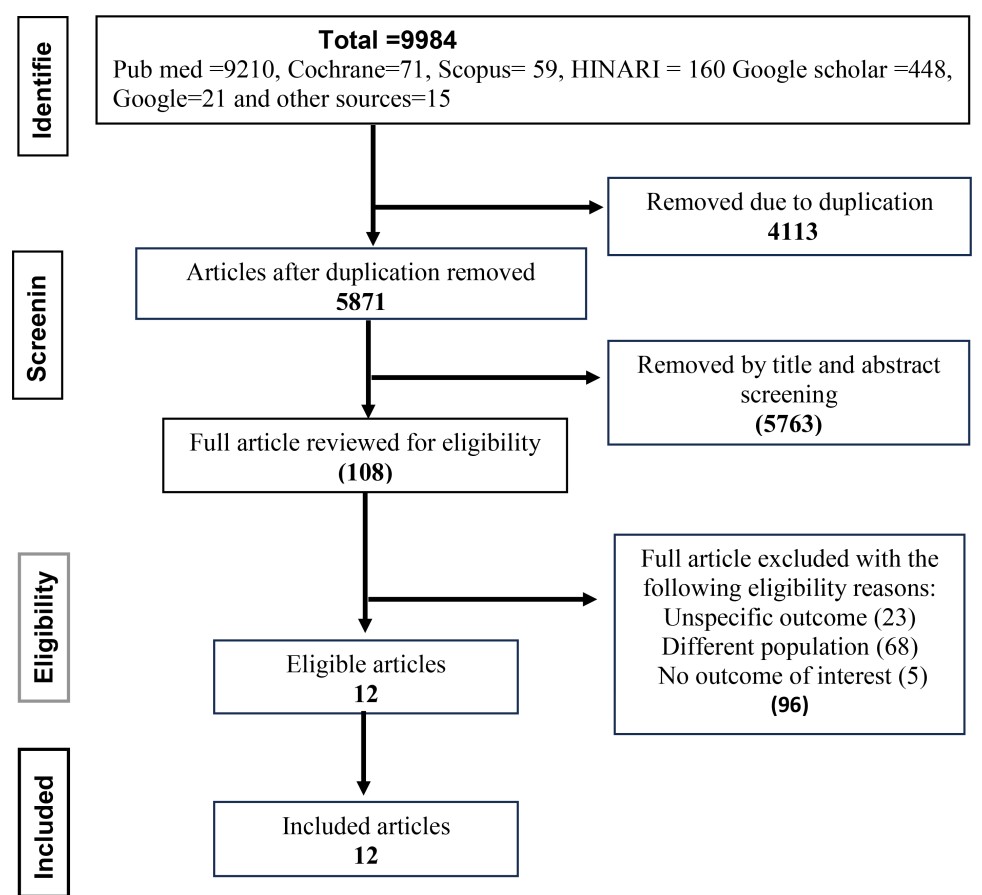

**Fig 1. 2020 PRISMA flow chart illustrating the process of search and selection of studies, Ethiopia, 2024.**

**Table 1. Characteristics of studies included in the meta-analysis for the pooled prevalence of maternal complication of instrumental vaginal delivery, Ethiopia, 2024.**

| Authors | Year | Region | Setting | Participants | Design | Sample size | Quality |
|---|---|---|---|---|---|---|---|
| Yohannes A et al. [10] | 2021 | Oromia | Mix | Mothers deilverd by OVD | Crossectional | 406 | 6 |
| Abegizer A et al. [11] | 2015 | Oromia | General | Mothers deilverd by OVD | Crossectional | 235 | 6 |
| Biru S et al. [3] | 2019 | Amhara | Tertiary | Mothers deilverd by OVD | Crossectional | 397 | 8 |
| Gebre S et al. [19] | 2017 | Tigray | General | Mothers deilverd by OVD | Crossectional | 357 | 7 |
| Hubena Z et al. [14] | 2018 | Oromia | Tertiary | Mothers deilverd by OVD | Crossectional | 242 | 6 |
| Israel E et al. [15] | 2023 | SNNP | Mix | Mothers deilverd by OVD | Crossectional | 399 | 8 |
| Sewunet H et al. [12] | 2022 | Amhara | Mix | Mothers deilverd by OVD | Crossectional | 313 | 8 |
| Shaka M et al. [2] | 2019 | SNNP | Tertiary | Mothers deilverd by OVD | Crossectional | 209 | 7 |
| Shimalis C et al. [8] | 2022 | Oromia | Mix | Mothers deilverd by OVD | Crossectional | 282 | 8 |
| Sium A et al. [13] | 2022 | Addis Ababa | Tertiary | Mothers deilverd by OVD | Crossectional | 238 | 7 |
| Nademo S et al. [18] | 2020 | SNNP | General | Mothers deilverd by OVD | Crossectional | 326 | 8 |
| Kassahun G et al. [20] | 2024 | Sidama | General | Mothers deilverd by OVD | Crossectional | 341 | 8 |

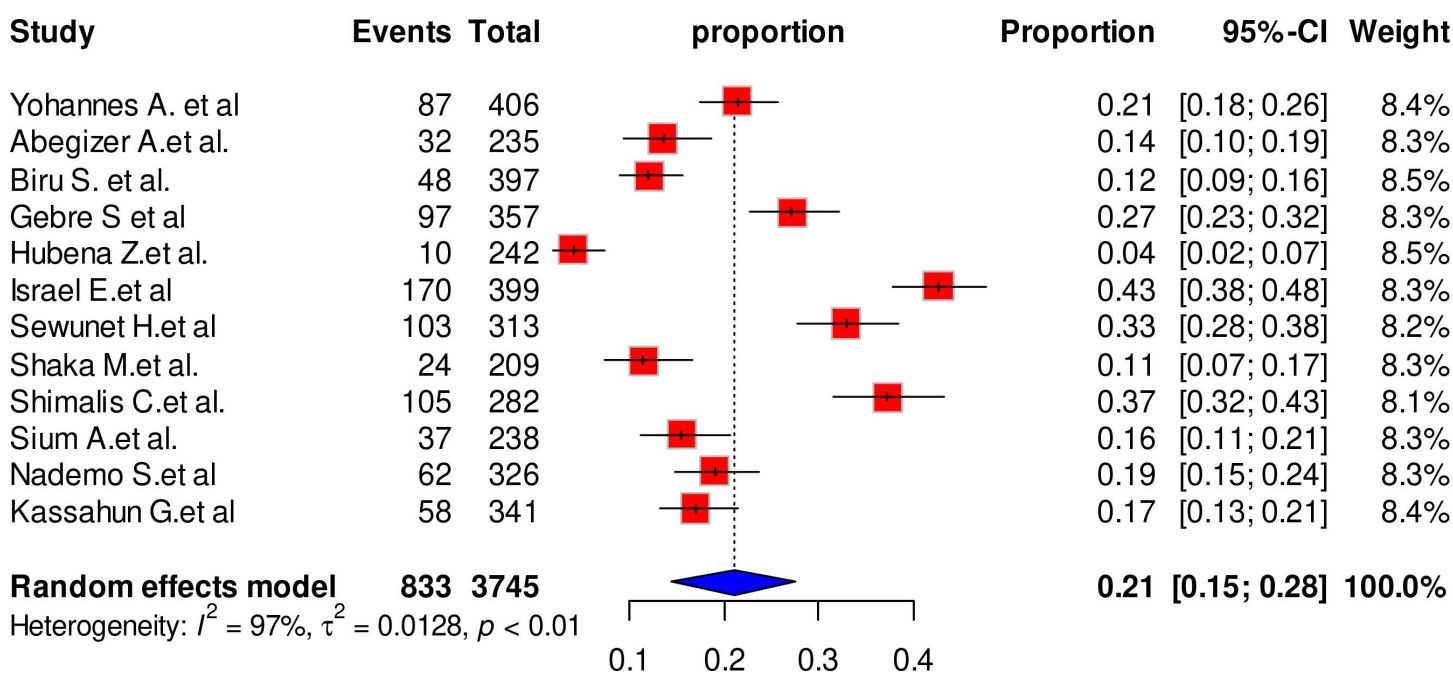

**Fig 2. Forest plot showing the pooled prevalence of maternal complication related to IVD, Ethiopia, 2024.**

## Sub-group analysis

The Pooled prevalence of maternal complication related IVD was further compared by region, setting and year of publication.

The subgroup analysis revealed that year of publication significantly modified overall prevalence of maternal complication related to IVD. Maternal complication related to IVD was lowest (14%) in studies conducted before 2020 and highest (26%) among studies conducted 2020 and after (P-value <0.002). However, the heterogeneity among the articles within each of these subgroups are very high (before 2020 = 95%; 2020 and after= 95%) (Fig 3).

Another subgroup analysis indicated statically significant difference in maternal complication between studies conducted at tertiary hospitals and those conducted mixed settings. The highest prevalence (33%) was reported from studies conducted at mixed facilities and the lowest prevalence (11%) was reported from studies conducted at tertiary hospitals (P-value = 0.98) but the heterogeneity remained very high (mixed: I² = 94%, general hospital: I² = 84%, tertiary hospital: I² = 84% (Fig 4).

Subgroup analysis by region showed no statistically significant difference (p= 0.98), with heterogeneity exceeding 92% in both subgroups (Fig 5).

## Meta regression and sensitivity analysis

We used multivariate meta-regression to assess how study characteristics affected the pooled estimates. None of the study characteristics (sample size, year of publication and study quality) were significantly associated with pooled prevalence of maternal complication related to IVD (Table 2).

Furthermore, a sensitivity analysis was carried out to evaluate the effect of a single study on the pooled estimate. The sensitivity analysis revealed that no single study significantly impacted the overall prevalence of maternal complication related to IVD (Fig 6).

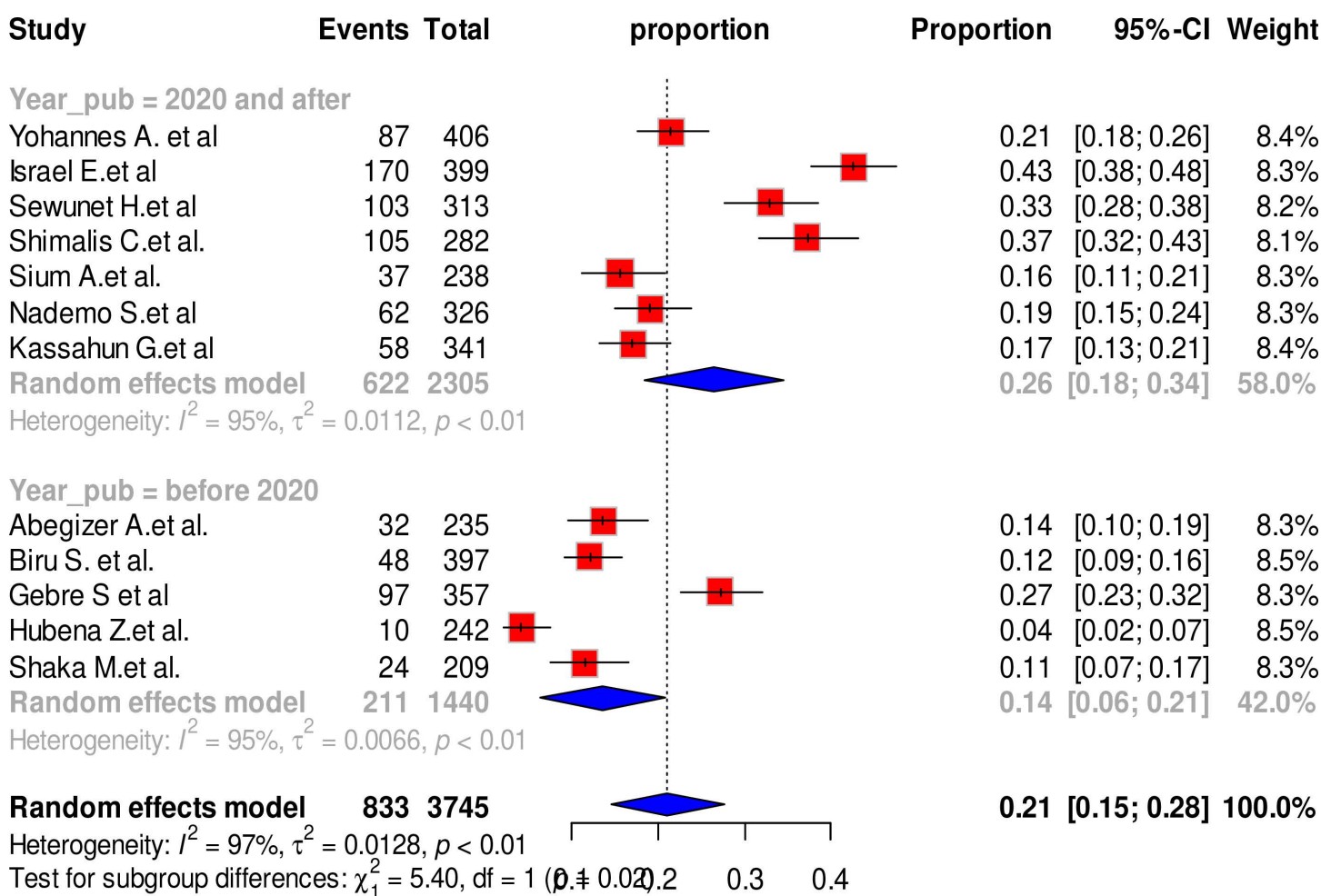

**Fig 3. Forest plot showing the pooled prevalence of maternal complication related to IVD by year of publication, Ethiopia, 2024.**

## Publication bias

The funnel plot showed an asymmetrical distribution of studies from the line of effect (Fig 7).

The Egger's test also showed a statistically significant publication bias with $B_0 = 0.08$, p-value = 0.023 (Fig 8).

Due to the presence of statically significant publication bias, a trim and fill analysis was performed. Hence, six studies were filled and the pooled prevalence of maternal complication related to IVD was adjusted to be 11% (95% CI: 3.0; 19.0) (Fig 9).

## Factors associated with maternal complication of IVD

Among the 12 studies analyzed to determine the pooled maternal complications related to IVD, 7 were excluded from the factor analysis because they did not assess factors or the factors were inadequate for meta-analysis.

Among those included studies, 4 variables (type of instrument, episiotomy status, parity and birth weight) were found significantly associated with pooled prevalence of maternal complication related to IVD (Table 3).

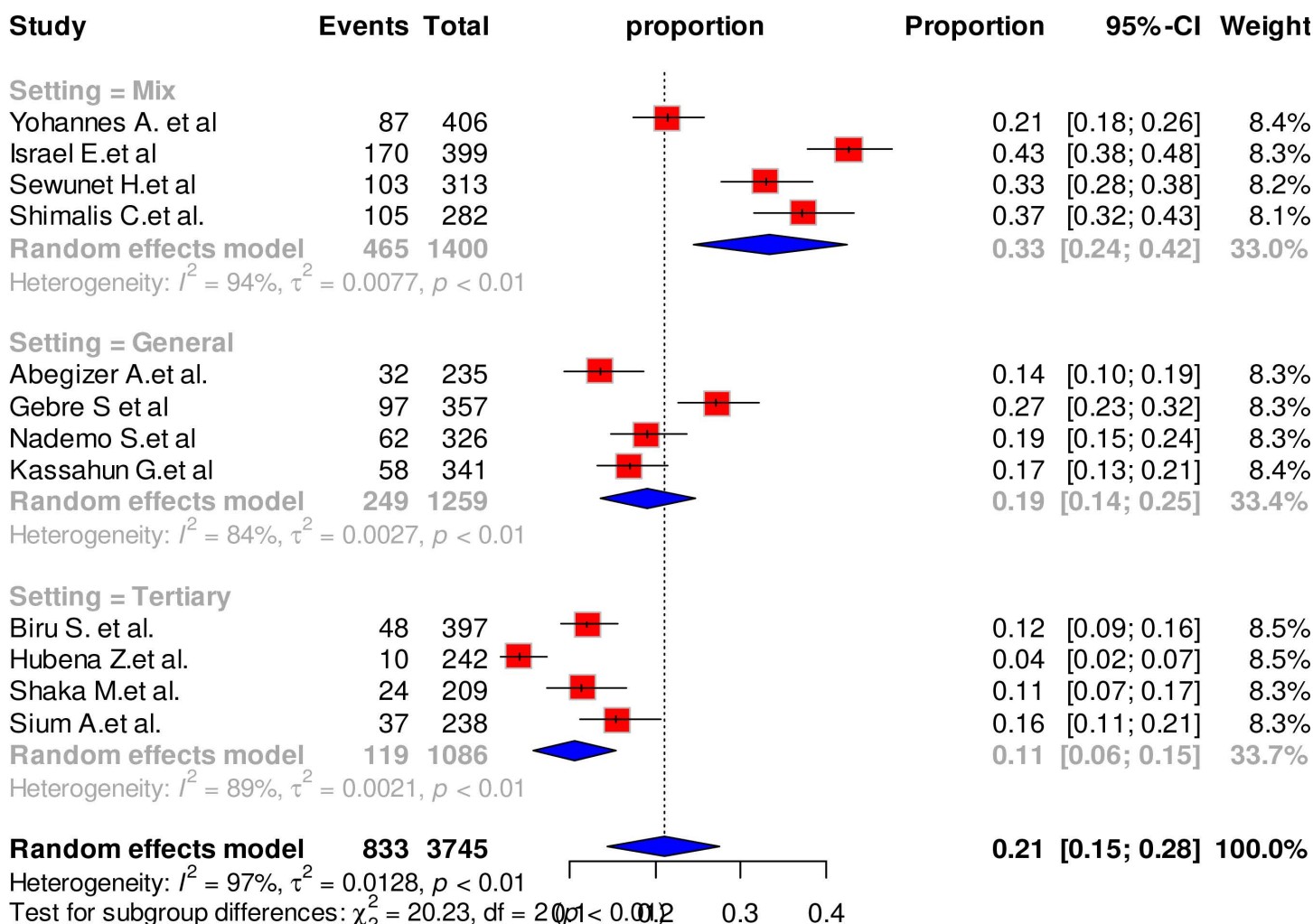

**Fig 4. Forest plot showing the pooled prevalence of maternal complication related to IVD by study setting, Ethiopia, 2024.**

**Type of instrument.** Three studies [3,18,20], involving 1064 participants, evaluated this content. The random effect model revealed that type of the instrument used to attend the childbirth (forceps vs vacuum) was significantly associated with the pooled estimate. The odds of having maternal complication related to IVD was higher among mothers who gave birth through forceps compared to vacuum extractor (OR = 1.99, 95%CI = 1.37; 2.90, $I^2$ = 0.0%, $p$ = 0.0003) (Table 3).

**Episiotomy.** The effect of episiotomy status was assessed with two studies [3,20] involving 723 participants. According to the random effect model result, episiotomy status was identified as an important modifier of the pooled estimate. The analysis indicated an increased odds of maternal complication related to IVD when the instrument is applied before performing an episiotomy (OR = 3.49, 95%CI = 2.12; 5.76, $I^2$ = 0.0%, $p$ = 0.0001) (Table 3).

**Birth weight.** The effect of birthweight was assessed with three studies [14,18,20] involving 909 participants. Birth weight was identified as another significant factor associated with the pooled estimate maternal complication related to IVD. Mothers who gave birth to baby weighing more than 4 kg hade higher odds of maternal complication related to IVD

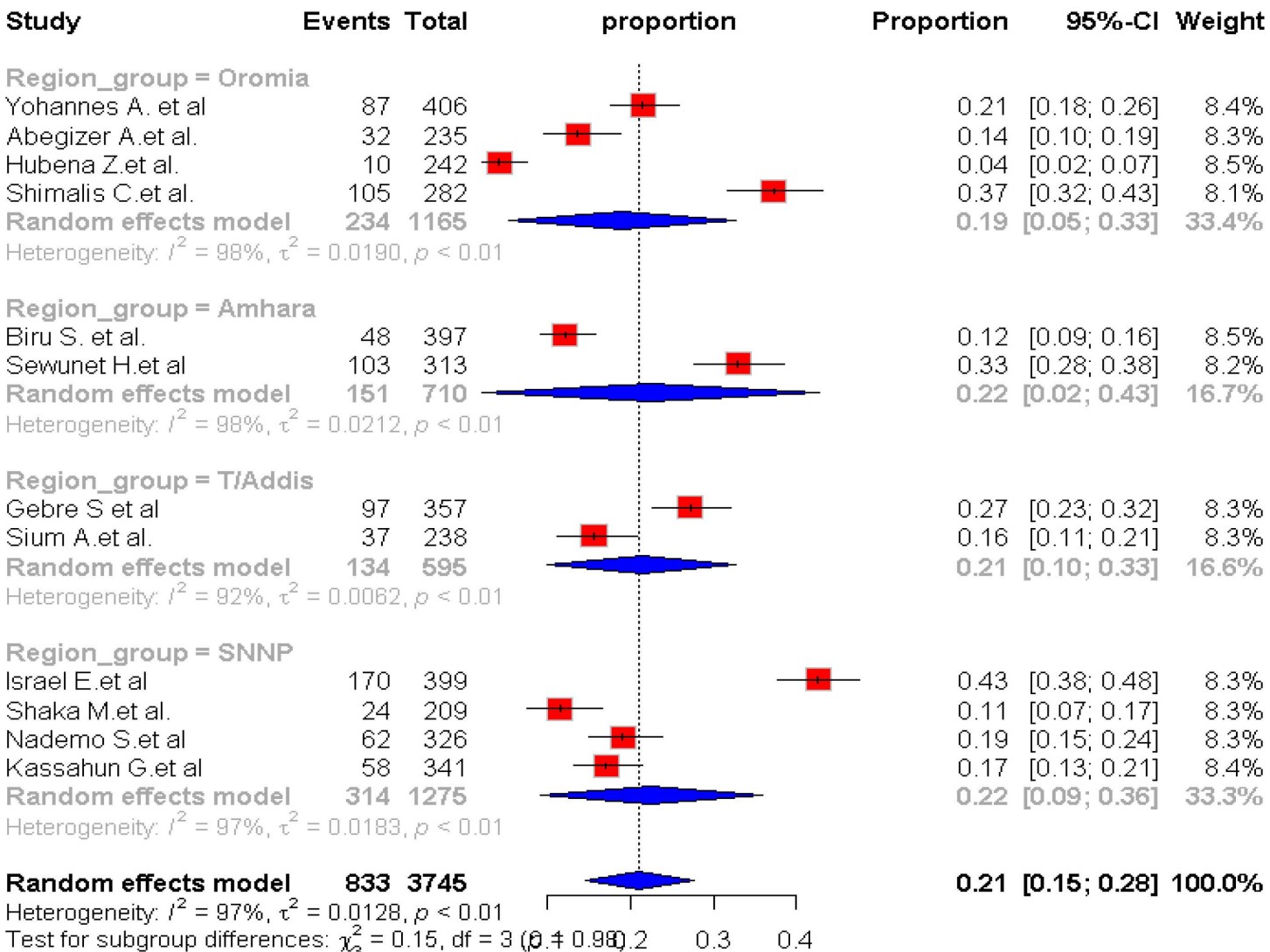

**Fig 5. Forest plot showing the pooled prevalence of maternal complication of IVD by region of publication, Ethiopia, 2024.**

**Table 2. Meta-regression analysis using publication year and sample size for the possible source of heterogeneity of maternal complication related to IVD, Ethiopia, 2024.**

| Variables | Coefficient | Std. Error | P-value |
|---|---|---|---|
| Sample size of the publication | -0.0001. | 0.001 | 0.123 |
| Year of publication | 0.0212 | 0.012 | 0.107 |
| Publication quality | 0.1701 | 0.108 | 0.155 |

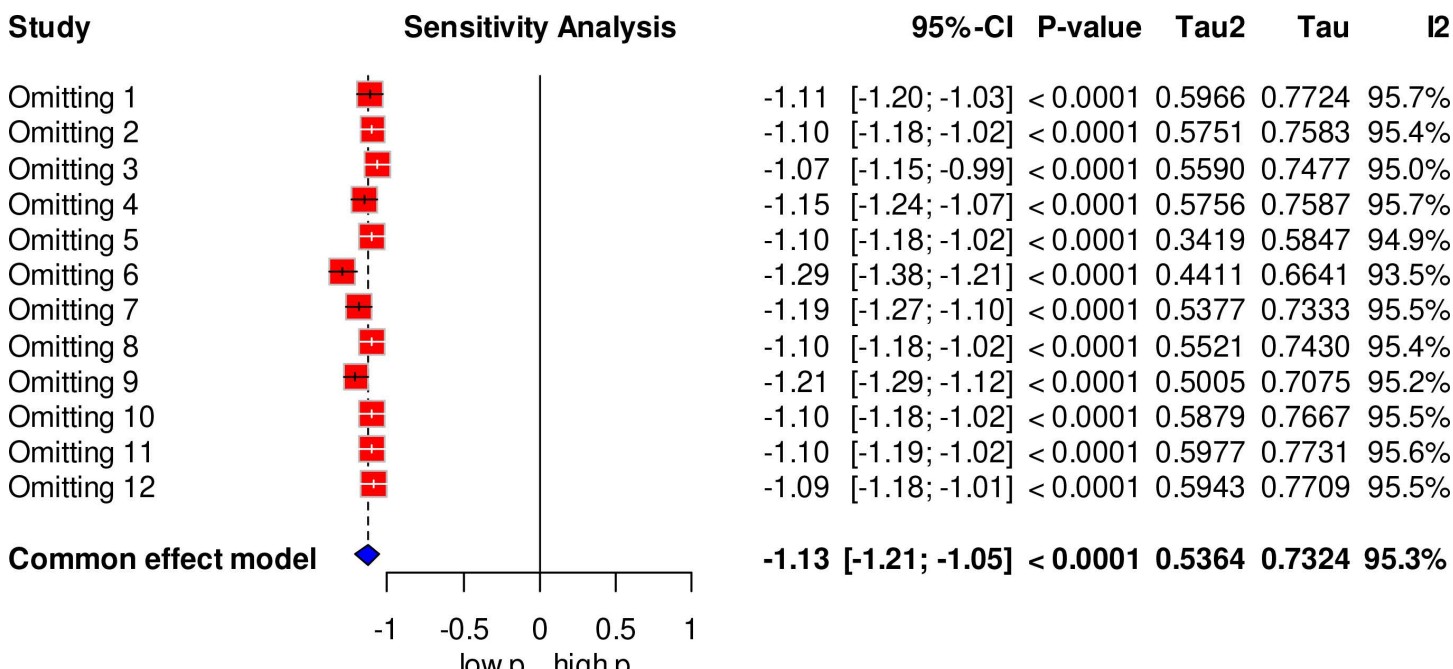

**Fig 6. Forest plot showing the sensitivity analysis of maternal complication related to IVD, Ethiopia, 2024.**

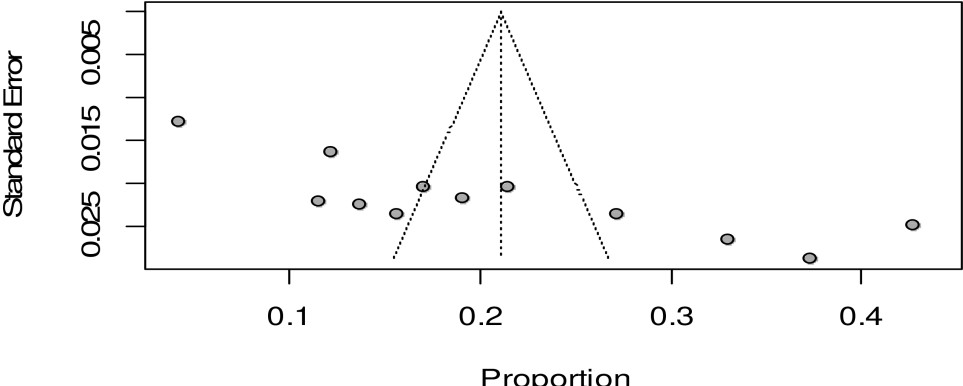

**Fig 7. Funnel plot showing publication bias among studies used to compute the pooled prevalence of maternal complication related to IVD, Ethiopia, 2024.**

compared to those who gave birth to baby weighing less than 4 kg (OR = 3.06, 95%CI = 1.88; 4.97, $I^2$ = 0.0%, $p$ = 0.0001) (Table 3).

**Parity.** Two studies [3,20] with a total participant of 738, assessed this content. The model showed that parity was significantly associated with the pooled estimate. Primiparous mothers were with increased odds of instrumental complication when compared to multiparous mothers (OR = 2.96, 95%CI = 1.80; 4.85, $I^2$ = 0.0%, $p$ = 0.0001) (Table 3).

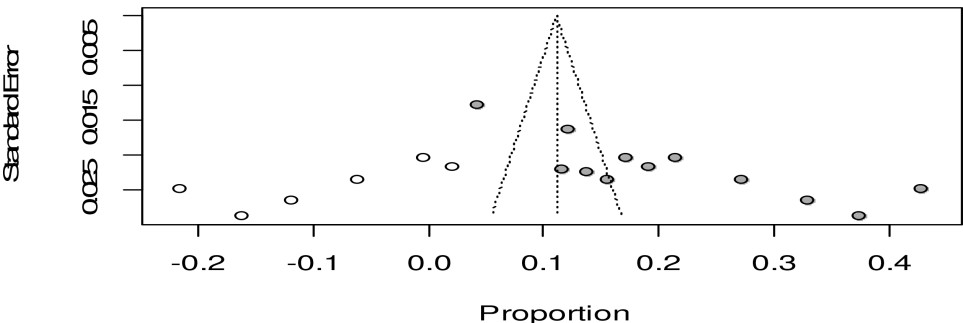

**Fig 8. Forest plot showing the trim fill analysis for the pooled prevalence of maternal complication related to IVD, Ethiopia, 2024.**

| Study | PRAW | SE(PRAW) | Proportion | Proportion | 95%-CI | Weight |
|---|---|---|---|---|---|---|
| Yohannes A. et al | 0.2143 | 0.0204 | | 0.21 | [0.17; 0.25] | 5.6% |
| Abegizer A.et al. | 0.1362 | 0.0224 | | 0.14 | [0.09; 0.18] | 5.6% |
| Biru S. et al. | 0.1209 | 0.0164 | | 0.12 | [0.09; 0.15] | 5.6% |
| Gebre S et al | 0.2717 | 0.0235 | | 0.27 | [0.23; 0.32] | 5.6% |
| Hubena Z.et al. | 0.0413 | 0.0128 | | 0.04 | [0.02; 0.07] | 5.6% |
| Israel E.et al | 0.4261 | 0.0248 | | 0.43 | [0.38; 0.47] | 5.5% |
| Sewunet H.et al | 0.3291 | 0.0266 | | 0.33 | [0.28; 0.38] | 5.5% |
| Shaka M.et al. | 0.1148 | 0.0221 | | 0.11 | [0.07; 0.16] | 5.6% |
| Shimalis C.et al. | 0.3723 | 0.0288 | | 0.37 | [0.32; 0.43] | 5.5% |
| Sium A.et al. | 0.1555 | 0.0235 | | 0.16 | [0.11; 0.20] | 5.6% |
| Nademo S.et al | 0.1902 | 0.0217 | | 0.19 | [0.15; 0.23] | 5.6% |
| Kassahun G.et al | 0.1701 | 0.0203 | | 0.17 | [0.13; 0.21] | 5.6% |
| Filled: Nademo S.et al | 0.0200 | 0.0217 | | 0.02 | [0.00; 0.06] | 5.6% |
| Filled: Yohannes A. et al | -0.0041 | 0.0204 | | 0.00 | [0.00; 0.04] | 5.6% |
| Filled: Gebre S et al | -0.0615 | 0.0235 | | 0.00 | [0.00; 0.00] | 5.6% |
| Filled: Sewunet H.et al | -0.1189 | 0.0266 | | 0.00 | [0.00; 0.00] | 5.5% |
| Filled: Shimalis C.et al. | -0.1622 | 0.0288 | | 0.00 | [0.00; 0.00] | 5.5% |
| Filled: Israel E.et al | -0.2159 | 0.0248 | | 0.00 | [0.00; 0.00] | 5.5% |
| **Random effects model** | | | | 0.11 | [0.03; 0.19] | 100.0% |

Heterogeneity: $I^2 = 98\%$, $\tau^2 = 0.0318$, $p < 0.01$

**Fig 9. Forest plot showing the trim fill analysis for the pooled prevalence of maternal complication related to IVD, Ethiopia, 2024.**

Table 3. **Meta-regression analysis predictors of maternal complication related to IVD, Ethiopia, 2024.**

| Factors | Number of studies | Total sample | Pooled OR (95% CI) | Heterogeneity I² (%) | P-value |
|---|---|---|---|---|---|
| Primiparous | 2 | 738 | 2.96 [1.80; 4.85] | 0.0% | <0.0001 |
| Forceps | 3 | 1064 | 1.99 (1.37; 2.90) | 0.0% | <0.0003 |
| Episiotomy | 2 | 738 | 5.18 [2.64; 10.15] | 0.0% | <0.0001 |
| Big birth weight | 3 | 909 | 3.06 [1.88; 4.97] | 0.0% | <0.0001 |

## Discussion

Maternal complication related to IVD is a very common condition, particularly in developing countries. Therefore, evidence regarding the burden of this problem and its risk factors is very crucial for policy makers in potentiating the efforts aimed at improving the poor maternal health in those countries. This study analyzed 12 original articles, encompassing 3745 mothers who have given birth through IVD.

This systematic review and meta-analysis revealed that the overall prevalence of maternal complications associated with IVD was 21.0% (95% CI: 0.15–0.28).

The test for subgroup differences indicated notable differences in effect size across subgroups. The year of publication and study setting significantly modified the estimate of overall maternal complication related to IVD. The pooled maternal complication related to IVD was higher among studies conducted 2020 and later (26%) compared to studies conducted before 2020 (14%). Similarly, the highest maternal complication related to IVD (33%) was reported among studies conducted at mixed facilities that included health centers and primary hospitals, while studies conducted at tertiary hospital reported lowest pooled maternal complication related to IVD (11%). The observed difference could be explained by the variation in human resources and infrastructure, as tertiary hospitals are equipped with well-trained providers and more advanced infrastructure, unlike health centers and primary hospitals. However, there is a substantially high heterogeneity within each of these subgroups. For studies conducted before 2020, the heterogeneity was extremely high (I² = 95%), and this level of heterogeneity persisted in studies conducted from 2020 onward (I² = 95%). Similarly, heterogeneity was notable among studies conducted in mixed settings, including health centers and primary hospitals (I² = 94%), as well as those conducted in tertiary hospitals (I² = 89%).

This high level of heterogeneity remained unexplained, suggesting that factors contributing to the variability in the study outcomes were either not accounted for or not adequately analyzed. Therefore, any interpretations or conclusions drawn from these subgroup analyses should be approached with caution, taking into consideration the potential influence of unmeasured confounders and variability in study methodologies [21]. Furthermore, in this systematic review and meta-analysis, we identified significant publication bias, indicating that the original estimate might have been influenced by the underreporting of studies with lower complication rates. After adjusting for small-study effects using the trim-and-fill method, the pooled prevalence was revised to 11% (95% CI: 3.0–19.0)

On the other hand, the current finding contradicts the result of prior systematic review and meta-analysis conducted in Ethiopia, which reported a prevalence of 11% for maternal complication related to IVD [6].

The possible explanation for the observed difference could be attributed to variations in the measurements of maternal complication related to IVD. The previous study assessed maternal complications solely in terms of perineal tear, whereas the current study considered more than seven types of complications.

Numerous studies have identified various factors affecting proportion of maternal complication related to IVD. The type of instrument, episiotomy, parity and birthweight were among the most significant factors [3,14,18,20].

It was identified that type of the instrument used is one of the predictors of maternal complication related to IVD. A study conducted by Biru S et al. found that maternal complication was more common with forceps deliveries than vacuum deliveries [3]. Similarly, a study conducted by Nademo S et al. in the Sidama Zone revealed that the use of forceps device was significantly associated with an unfavorable maternal outcome [18]. Furthermore, a recent study by Kasahun G et al. documented that forceps assisted vaginal delivery was more likely to result in immediate adverse maternal outcomes compared to vacuum-assisted vaginal delivery [20].

However, studies conducted by Huben Z et al. at Jimma Medical Centre and by Hailu S et al. at Suhul Hospital reported there is no difference in maternal complication whether forceps or vacuum is used [14,19]. Similarly, a study by Sium et al. also reported no significant association between the type of instrument and maternal complication [13]. This variation could be attributed to the practice of instrumental shifting practice in the later studies, which makes it difficult to determine whether the complications are attributable to forceps or vacuum use.

Several studies have shown that the birth weight of the newborn is an important factor in predicting maternal complication related to IVD. According a study conducted by Kasahun G et al., Mothers who gave birth to a neonate with a birth weight ≥4000 grams were more likely to experience immediate maternal complication related to IVD compared to those who gave birth to a neonate weighing <4000 grams [20].The studies conducted at Jimma University medical center and Nigist Elleni Memorial comprehensive hospital revealed that those mothers who gave birth to neonate weighing ≥4000 grams are less likely to have favorable maternal outcome compared with neonates of normal birth weight [14,18]. This may be because macrosomia contributes to uterine atony and perineal lacerations, which can lead to postpartum hemorrhage [22,23]. On the other hand, it can be understood that instruments are being applied in cases of contraindicated labor, as macrosomia is a contraindication for instrumental vaginal delivery [24]. Therefore, having clearly estimated fetal weight helps to avoid contraindicated instrumental application and reduces the complication of IVD.

Episiotomy is also another important factor affecting the overall maternal complication associated to IVD. A study conducted at Felege Hiwot Hospital in Amhara region by Biru S et al. reported that mothers who had an episiotomy were at lower risk for maternal complication during instrumental delivery compared to those who did not have an episiotomy [3]. However, in contrast, a study by Kasahun G et al. found that mothers who underwent an episiotomy during instrumental delivery were more likely to experience maternal complication compared to those who did not [20].

This contradicting result might be due to difference in skill and techniques of episiotomy. If adequately sized episiotomy is not performed during instrumental delivery, there is a risk of tearing into the rectal tissues and anal sphincter. Additionally, type of episiotomy also critical. During instrumental vaginal delivery, mediolateral or lateral episiotomy has been shown to significantly reduces the risk of anal sphincter injuries [25].

## Limitation

Ethiopia has more than 15 regional states, but only 6 of them represented in this systematic review and meta-analysis. More than 10 variables that were found to be significant in original studies were excluded from, the factor analysis because it reported in only one primary article. Moreover, minor modifications were made to the original classifications to ensure compatibility among the variables included in the analysis. Above all, there is significantly high and unexplained heterogeneity; therefore, the results should be interpreted with caution.

## Conclusion

This systematic review and meta-analysis revealed the pooled prevalence of maternal complication related to instrumental vaginal delivery in Ethiopia is high. Type of the instrument used, parity, duration of labor, and birth weight were potential predictors of maternal complication of instrumental vaginal delivery. Effective evaluation of indication, contra indication and precondition for each instrument helps to prevent the maternal complication of instrumental vaginal delivery.

## Supporting information

**S1 File. All accessed literatures.**
(XLSX)

**S2 File. Quality appresal.**
(XLS)

**S3 File. Extracted data.**
(XLSX)

**S1 Checklist. PRISMA 2020 checklist.**
(DOCX)

## Author contributions

**Conceptualization:** Beyene Sisay Damtew, Teresa Kisi Beyen.

**Data curation:** Elias Bekele Wakwaya, Beker Ahmed Hussen, Teresa Kisi Beyen.

**Formal analysis:** Hinsermu Bayu Abdi, Teresa Kisi Beyen.

**Investigation:** Elias Bekele Wakwaya.

**Methodology:** Hinsermu Bayu Abdi, Beker Ahmed Hussen, Teresa Kisi Beyen.

**Software:** Hinsermu Bayu Abdi.

**Writing – original draft:** Hinsermu Bayu Abdi, Beyene Sisay Damtew.

**Writing – review & editing:** Hinsermu Bayu Abdi, Elias Bekele Wakwaya, Teresa Kisi Beyen.

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
