## [Decision Letter · Decision Letter 0]

5 Nov 2024

PONE-D-24-44325Maternal Complication of Instrumental Vaginal Delivery and Its Associated Factors among Mothers Who Underwent Instrumental Vaginal Delivery: Systematic Review and MetanalysisPLOS ONE

Dear Dr. Abdi,

Thank you for submitting your manuscript to PLOS ONE. After careful consideration, we feel that it has merit but does not fully meet PLOS ONE’s publication criteria as it currently stands. Therefore, we invite you to submit a revised version of the manuscript that addresses the points raised during the review process.

 Please submit your revised manuscript by Dec 20 2024 11:59PM. If you will need more time than this to complete your revisions, please reply to this message or contact the journal office at plosone@plos.org . Please include the following items when submitting your revised manuscript:

We look forward to receiving your revised manuscript.

Kind regards,

Liknaw Bewket Zeleke, MSc

Academic Editor

PLOS ONE

2. As required by our policy on Data Availability, please ensure your manuscript or supplementary information includes the following:

Additional Editor Comments (if provided):

Reviewers' comments:

Reviewer's Responses to Questions

**Comments to the Author**

1. Is the manuscript technically sound, and do the data support the conclusions?

Reviewer #1: Yes

Reviewer #2: Yes

2. Has the statistical analysis been performed appropriately and rigorously?

Reviewer #1: Yes

Reviewer #2: Yes

3. Have the authors made all data underlying the findings in their manuscript fully available?

Reviewer #1: Yes

Reviewer #2: Yes

4. Is the manuscript presented in an intelligible fashion and written in standard English?

Reviewer #1: Yes

Reviewer #2: No

5. Review Comments to the Author

Reviewer #1: Thank you for the opportunity to review this manuscript titled "Maternal Complication of Instrumental Vaginal Delivery and Its Associated Factors among Mothers Who Underwent Instrumental Vaginal Delivery: Systematic Review and Meta-analysis".

This study is both timely and significant, as it addresses the important issue of maternal complications associated with instrumental vaginal delivery (IVD), a procedure commonly used in obstetric care, especially in developing countries such as Ethiopia. The sample used for this meta-analysis included 12 studies with a total of 3,745 participants, providing a substantial foundation to estimate the pooled prevalence of maternal complications related to IVD.

Research like this, focusing on developing countries, is essential because it brings to light context-specific challenges in maternal and child health. In these settings, health systems are often under-resourced, and understanding the factors contributing to maternal morbidity is key to improving outcomes. This study’s meta-analytical approach, which pools data from various studies, provides a more comprehensive and accurate picture of the national burden of IVD complications, as opposed to isolated studies. It also allows for the identification of common risk factors, thus guiding targeted interventions to reduce maternal complications.

1. Significance and Novelty:

The paper provides a comprehensive overview of the prevalence and predictors of maternal complications associated with instrumental vaginal delivery (IVD) in Ethiopia. The authors' effort to systematically review and meta-analyze studies conducted in this context is commendable, as it addresses a critical gap in the obstetric care literature in Ethiopia and potentially guides future clinical practices and policies. The novelty of the paper lies in its focus on a specific geographical location (Ethiopia) and its attempt to collate and analyze data from various studies to provide a national-level perspective on IVD complications, which has not been extensively covered in previous literature.

2. Strengths:

The methodology is robust, employing a comprehensive search strategy across multiple databases and including unpublished literature to minimize publication bias. This approach enhances the reliability of the findings.

The inclusion of a wide range of predictors for maternal complications (type of instrument, episiotomy status, birthweight, and parity) provides valuable insights into the factors that could be addressed to mitigate risks associated with IVD.

The statistical analysis, including subgroup analysis and meta-regression, is thorough, allowing for a nuanced understanding of the data.

The paper is of high relevance to obstetric care in Ethiopia and other similar contexts, offering evidence that can inform practice, policy, and future research.

3. Weaknesses:

The heterogeneity among included studies is very high (I² = 97%), which suggests that the findings might be influenced by underlying variability among the study designs, populations, and contexts. This issue is acknowledged but not sufficiently addressed in the analysis and discussion sections.

The paper lacks a clear discussion of the limitations of the included studies and how they might impact the overall findings and conclusions.

Some important predictors of maternal complications in IVD, such as the operator's experience or the healthcare facility's level (beyond tertiary and general hospitals), are not explored in depth.

The paper does not sufficiently engage with existing literature on the subject, missing an opportunity to contextualize its findings within the broader global research landscape on instrumental vaginal delivery.

Although the study presents the prevalence of complications, it would benefit from a more detailed discussion of the local health system context and cultural factors that might influence the practice of IVD and its outcomes in Ethiopia.

4. Suggestions for Improvement:

The authors could further explore and discuss the implications of the high heterogeneity among studies. Considering meta-regression or sensitivity analysis to identify potential sources of this heterogeneity could strengthen the paper.

Expanding the discussion section to more critically engage with the limitations of the included studies and their potential impact on the review's conclusions would provide a more balanced view of the findings.

Including additional predictors or conducting a more detailed analysis of the available predictors (e.g., operator experience, healthcare facility level) could enrich the analysis. If data availability is a limitation, this should be explicitly stated as an area for future research.

Enhancing the literature review to include a broader range of studies from different geographical and socio-economic contexts could help position the findings within the global discourse on maternal health and IVD. This would also allow the authors to more explicitly highlight the unique contributions of their study.

Further elaboration on strategies for minimizing IVD-related complications in low-resource settings, based on the study’s findings, would strengthen the conclusion and offer actionable recommendations for healthcare providers and policymakers.

Summary Statement:

This paper makes a valuable contribution to understanding the prevalence and predictors of maternal complications in instrumental vaginal delivery in Ethiopia. Its strengths lie in the comprehensive methodology and the relevance of its findings to obstetric care practices and policies. However, addressing the highlighted concerns regarding heterogeneity, discussion of limitations, exploration of additional predictors, and engagement with existing literature could significantly enhance the paper's impact and relevance. With these improvements, the paper has the potential to be an important resource for clinicians, policymakers, and researchers interested in maternal health in Ethiopia and similar contexts.

Reviewer #2: Maternal Complication of Instrumental Vaginal Delivery and Its Associated Factors among Mothers Who Underwent Instrumental Vaginal Delivery: Systematic Review and Metanalysis

Summary of the Research

This study aimed to assess pooled prevalence of maternal complication of instrumental vaginal delivery and its associated factors among mothers who underwent instrumental vaginal delivery. Data based: PubMed, Scopus, HINARI, Cochrane library, and Google Scholar databases for studies published until August 2024. Keywords such as instrumental delivery, forceps, vacuum, complication, factors and Ethiopia were used to access literatures from the databases. For quality assessment and data extraction, the Joanna Briggs Critical Appraisal Tools and the Preferred Reporting Items for Systematic Review and Meta-Analyses were utilized. Random-effect model was used to calculate the pooled prevalence of maternal complication of instrumental vaginal delivery. Subgroup analyses were also conducted to explore potential heterogeneity. The publication bias was assessed using Funnel plot and Egger’s. A total of 12 studies with 3745 study participants were participated in the present meta-analysis. The pooled prevalence of maternal complication of instrumental vaginal delivery was 21% (95% CI, 15.0%-28.0%). The prevalence of maternal complication was significantly different among studies conducted between [2015 - 2020), and those conducted between [2020 and 2024]. Lowest (14%) in studies conducted before 2020 and highest (26%) among studies conducted 2020 and after (P-value<0.001).

Type of instrument (AOR: 1.99, 95% CI: 1.37–2.90), episiotomy status (AOR: 3.49, 95% CI: 2.12– 5.76), birthweight (AOR: 3.06, 95% CI: 1.88 – 4.97) and parity (AOR=2.96,95%CI:1.80 – 4.85) were the factors associated with maternal complication of instrumental vaginal delivery. This study shows that approximately one in five mothers who underwent instrumental vaginal delivery develop serious maternal complication. Type of the instrument, episiotomy status, birthweight and parity were important predictors of the maternal complication of instrumental vaginal delivery. Effective evaluation of indication, contra indication and precondition for each instrument helps to prevent the maternal complication of instrumental vaginal delivery.

Areas for improvement

Title:

The authors wrote the title in a good way and reflecting the aim and objectives of the Systematic review and meta-analysis. However, the authors should make the title shorter.

Abstract:

Abstract is well written.

Please, don’t use abbreviation in short title.

The authors should mention clearly the main objective of the systematic review and meta-analysis.

Correct all metanalysis with meta-analysis

Remove please the word “Systematic review and meta-analysis” from the key word and remove “pooled” prevalence is enough. Add “factors” to the key word

Make sure to add key word in page no 3 after abstract.

Introduction Section:

In the first paragraph,

Please don’t repeat the full form of IVD. First time is enough then you can use only abbreviation “IVD”.

The authors should make sure to identify the importance of this SR and MA as it is not only to pooled the complication but also the prevalence. Be consistence please.

The authors should revise introduction section for grammar issues and language to improve readability.

The authors should indicate what they wanted the readers to understand.

Overall

The authors need to arrange introduction section. Rewrite some paraphrase so, the readers will understand what you wanted to convey from messages. They also need to make sure to clearly mention the significant of the systematic review and meta-analysis and the aim of it.

Material and Methods:

Method,

Clear.

Study protocol and registration,

The authors should make sure that the PRISMA flow diagram is appeared in this section.

The authors should write in brackets synonyms, terms and keywords used for this systematic review.

The authors should write examples of how they enriched search using Boolean operators and phrase search.

The authors should write the reason for selecting grey literature.

The authors should mention how many articles excluded for this systematic review, then provide the reasons.

Quality Assessment,

The authors should explain the reason for selecting JBI and its importance in more detail.

The authors should explained the numbers of designs in this systematic review, how many quantitative studies with its design and how many qualitative with its design.

Statistical analysis,

Clear.

Results:

Well explained results section however, the authors should revise the language and grammar to improve readability.

Studies characteristics, clear.

The authors should include figures and tables within the results section.

Discussion:

The authors should discuss more about this results from different angles.

The authors should discuss in more detail about the studies in related to prevalence and factors.

The authors should make clear for the readers about the themes that they were discussing.

The authors should discuss each themes from different angle and in detail.

The authors should revise the language and grammar to improve readability.

References:

The authors should revise all references according to the guidelines provided.

6. PLOS authors have the option to publish the peer review history of their article (what does this mean? ). If published, this will include your full peer review and any attached files.

**Do you want your identity to be public for this peer review?** For information about this choice, including consent withdrawal, please see our Privacy Policy .

Reviewer #1: **Yes: ** Olivier Mukuku

Reviewer #2: **Yes: ** Zalikha Al-Marzouqi

---

## [Author Response · Author response to Decision Letter 1]

13 Jan 2025

Response to Reviewers

Reviewer 1

1. Please ensure that your manuscript meets PLOS ONE's style requirements, including those for file naming. The PLOS ONE style templates can be found at ….

Authors’ response: The manuscript is edited according the journal guideline

2. As required by our policy on Data Availability, please ensure your manuscript or supplementary information includes the following:

A numbered table of all studies identified in the literature search, including those that were excluded from the analyses. For every excluded study, the table should list the reason(s) for exclusion

Authors response:

We agree that transparency in study selection is critical for the robustness of systematic reviews and meta-analyses. However, we respectfully disagree with the necessity of providing a table listing all identified studies, including excluded ones, for the feasibility reason.

According to the PRISMA 2020 checklist, we have used a flowchart (Figure 1, page 11, line 7) to illustrate the total number of studies identified during the literature search, including those excluded due to duplication (4113), title and abstract screening (5531), unspecific outcomes (120), different populations (165), or unrelated areas (44). The flowchart provides a clear summary of the study selection process and the reasons for exclusion at each stage. Creating a separate table for all identified studies, including those excluded, would be impractical given the large number of studies screened (over 9,000). We believe the flowchart serves as an appropriate and efficient alternative for documenting the study selection process, in alignment with PRISMA guidelines.

We used a numbered table (Table 1, page 12, line 1) to present detailed information about those of studies included in the final analysis.

Authors response:

We do have only one unpublished paper, M.Sc. student thesis, which is available online and reference number 10 is modified to include its URL to access the paper. See reference number 10, page 22 line 17 - 20

Authors response:

The requested information has already been included in the manuscript. We extracted data using the standard Excel format provided by the Joanna Briggs Institute's Meta-Analysis of Statistics Assessment and Review Instruments. The entire data extraction process was guided by this predetermined Excel format. For details, please refer to page 28, lines 1.

5. If applicable for your analysis, a table showing the completed risk of bias and quality/certainty assessments for each study or outcome. Please ensure this is provided for each domain or parameter assessed. For example, if you used the Cochrane risk-of-bias tool for randomized trials, provide answers to each of the signaling questions for each study. If you used GRADE to assess certainty of evidence, provide judgements about each of the quality of evidence factor. This should be provided for each outcome.

Authors’ response:

We assessed publication bias using the standard approach, including the funnel plot and Egger's test, and the results are already presented. See page 13, lines 16 - 22. Regarding the quality assessment, we used the JBI Critical Appraisal Tool for cross-sectional studies, and the overall quality of each paper was reported as one of the study characteristics and incorporated into Table 1, page 12, line 1.

6. An explanation of how missing data were handled. This information can be included in the main text, supplementary information, or relevant data repository. Please note that providing these underlying

Authors’ response:

The following statement has been incorporated under the results section: Among the twelve studies analyzed to determine the pooled maternal complications related to IVD, seven were excluded from the factor analysis because they did not assess factors or the factors were inadequate for meta-analysis.

7. Suggestions for Improvement:

The authors could further explore and discuss the implications of the high heterogeneity among studies. Considering meta-regression or sensitivity analysis to identify potential sources of this heterogeneity could strengthen the paper.

Authors response

In the analysis part, we have already conducted meta-regression to see if the study characteristics such as sample size, year of publication and quality of the studies have affected the estimate. Additionally, sensitivity analysis was conducted using both subjective (funnel plot) and objective measurement (egger’s test) and the results were also reported. See page 13 line 16 – 22.

Regarding the implication of high heterogeneity, the following statement was incorporated into the discussion part. This high level of heterogeneity remained unexplained, suggesting that factors contributing to the variability in the study outcomes were either not accounted for or not adequately analyzed. Therefore, any interpretations or conclusions drawn from these subgroup analyses should be approached with caution, taking into consideration the potential influence of unmeasured confounders and variability in study methodologies. Furthermore, in this systematic review and meta-analysis, we identified significant publication bias, indicating that the original estimate might have been influenced by the underreporting of studies with lower complication rates. After adjusting for small-study effects using the trim-and-fill method, the pooled prevalence was revised to 11% (95% CI: 3.0–19.0).

8. Including additional predictors or conducting a more detailed analysis of the available predictors (e.g., operator experience, healthcare facility level) could enrich the analysis. If data availability is a limitation, this should be explicitly stated as an area for future research.

Authors response:

We, the authors, completely agree with the reviewer's insightful observation regarding the potential influence of the operator's experience and the health facility level as factors affecting the estimate. However, during our comprehensive literature review and data extraction process, we did not come across any primary studies that explicitly explored these variables as potential contributors to the observed outcomes.

9. Enhancing the literature review to include a broader range of studies from different geographical and socio-economic contexts could help position the findings within the global discourse on maternal health and IVD. This would also allow the authors to more explicitly highlight the unique contributions of their study.

Authors response:

We identified only one previous systematic review and meta-analysis, which we used to discuss the pooled prevalence (see page 10, line 20). Otherwise, this is the first systematic review and meta-analysis conducted both in Ethiopia and internationally on this topic. We believe it would be inappropriate to compare the findings of this systematic review and meta-analysis with those of individual primary studies.

10. Further elaboration on strategies for minimizing IVD-related complications in low-resource settings, based on the study’s findings, would strengthen the conclusion and offer actionable recommendations for healthcare providers and policymakers.

Authors response:

According to our findings, complications of IVD are influenced by factors such as birth weight, episiotomy, type of instrument, and parity. Regarding birth weight, the following statement has been incorporated into the discussion:

It can be understood that instruments are often applied in cases of contraindicated labor, as macrosomia is a contraindication for instrumental vaginal delivery. Therefore, having an accurately estimated fetal weight is crucial to avoid contraindicated instrumental applications and to reduce complications associated with IVD.

For other variables affecting maternal complications of IVD, our suggestions have already been included in the discussion section. For example, see page 19, lines 8–11, for the discussion related to episiotomy. See page 19 line 12 - 19.

Reviewer 2

1. Title:

The authors wrote the title in a good way and reflecting the aim and objectives of the Systematic review and meta-analysis. However, the authors should make the title shorter.

Authors response:

Seven words were removed and the title become “Maternal Complication of Instrumental Vaginal Delivery and Its Associated Factors: Systematic Review and Meta-analysis

2. Abstract:

Abstract is well written.

2.1. Please, don’t use abbreviation in short title.

Authors response:

We haven’t used any abbreviation, but HINARI, AOR, CI which are standard acronyms

2.2. The authors should mention clearly the main objective of the systematic review and meta-analysis.

Authors response:

This statement “. Therefore, this study aimed to assess pooled prevalence of maternal complication related to instrumental vaginal delivery and its associated factors among mothers who underwent instrumental vaginal delivery in Ethiopia”. was already mention in abstract. See page 2 line 5-8.

2.3. Correct all metanalysis with meta-analysis

Authors response:

Edited throughout the document

3. Remove please the word “Systematic review and meta-analysis” from the key word and remove “pooled” prevalence is enough. Add “factors” to the key word

Make sure to add key word in page no 3 after abstract.

Authors response:

Done

4. Introduction Section:

In the first paragraph,

4.1. Please don’t repeat the full form of IVD. First time is enough then you can use only abbreviation “IVD”.

Authors response:

The second and third IVD was removed

4.2. The authors should make sure to identify the importance of this SR and MA as it is not only to pooled the complication but also the prevalence. Be consistence please.

Authors response:

The statement was paraphrased

4.3. The authors should revise introduction section for grammar issues and language to improve readability.

The authors should indicate what they wanted the readers to understand.

Overall

The authors need to arrange introduction section. Rewrite some paraphrase so, the readers will understand what you wanted to convey from messages. They also need to make sure to clearly mention the significant of the systematic review and meta-analysis and the aim of it.

Authors response

Done

5. Material and Methods:

Method,

Clear.

Study protocol and registration,

5.1. The authors should make sure that the PRISMA flow diagram is appeared in this section.

Authors response:

Thank you for your suggestion. In accordance with the PRISMA 2020 checklist, the PRISMA flow diagram has been included in the results section of the manuscript. This placement aligns with standard guidelines, which recommend that the flow diagram be used to transparently depict the study selection process.

5.2. The authors should write in brackets synonyms, terms and keywords used for this systematic review.

Authors response:

Already done (see page 6-7 line 16 – 6)

5.3. The authors should write examples of how they enriched search using Boolean operators and phrase search.

Authors response:

Already done (see page 6-7 line 16 – 6)

5.4. The authors should write the reason for selecting grey literature.

Authors response:

This sentence was incorporated “Additionally, the Arsi University College of Health Sciences Library was searched for potential unpublished literature to reduce small-study effects (publication bias).”

5.5. The authors should mention how many articles excluded for this systematic review, then provide the reasons.

Authors response:

This was done under result in PRISMA flow diagram

5.6. Quality Assessment,

The authors should explain the reason for selecting JBI and its importance in more detail.

Authors response:

The following phrase is included to explain the reasons why JBI is used. JBI was selected to evaluate the study's methodological quality and determine how effectively it addressed potential biases in its design.

5.7. The authors should explain the numbers of designs in this systematic review, how many quantitative studies with its design and how many qualitative with its design.

Authors response:

It was already mentioned that all studies were quantitative with cross-sectional design.

5.8. Statistical analysis,

Clear.

6. Results:

6.1. Well explained results section however, the authors should revise the language and grammar to improve readability.

Authors response:

Done

6.2. Studies characteristics, clear. The authors should include figures and tables within the results section.

Authors response:

Done

7. Discussion:

7.1. The authors should discuss more about these results from different angles. The authors should discuss in more detail about the studies in related to prevalence and factors.

Authors response:

We got only one similar meta-analysis and that was used in discussing prevalence. otherwise, this meta-analysis is the first in its kind, we faced difficulty to the comparator. On the other hand, it is not fair to discuss the result of this study with other primary studies.

7.2. The authors should make clear for the readers about the themes that they were discussing.

The authors should discuss each themes from different angle and in detail.

The authors should revise the language and grammar to improve readability.

Authors response:

Done

7.3. References:

The authors should revise all references according to the guidelines provided.

Authors response:

Done

Thank you all very much!

---

## [Decision Letter · Decision Letter 1]

12 Feb 2025

Maternal Complication of Instrumental Vaginal Delivery and Its Associated Factors: Systematic Review and Meta-analysis

PONE-D-24-44325R1

Dear Dr. Abdi,

We’re pleased to inform you that your manuscript has been judged scientifically suitable for publication and will be formally accepted for publication once it meets all outstanding technical requirements.

Kind regards,

Liknaw Bewket Zeleke, MSc

Academic Editor

PLOS ONE

Additional Editor Comments (optional):

Reviewers' comments:

Reviewer's Responses to Questions

**Comments to the Author**

1. If the authors have adequately addressed your comments raised in a previous round of review and you feel that this manuscript is now acceptable for publication, you may indicate that here to bypass the “Comments to the Author” section, enter your conflict of interest statement in the “Confidential to Editor” section, and submit your "Accept" recommendation.

Reviewer #2: All comments have been addressed

2. Is the manuscript technically sound, and do the data support the conclusions?

Reviewer #2: Yes

3. Has the statistical analysis been performed appropriately and rigorously?

Reviewer #2: Yes

4. Have the authors made all data underlying the findings in their manuscript fully available?

Reviewer #2: Yes

5. Is the manuscript presented in an intelligible fashion and written in standard English?

Reviewer #2: Yes

6. Review Comments to the Author

Reviewer #2: all the best, the authors addressed all the comments and changed accordingly. They make a tremendous effort to improve the manuscript. Thank you

7. PLOS authors have the option to publish the peer review history of their article (what does this mean? ). If published, this will include your full peer review and any attached files.

**Do you want your identity to be public for this peer review?** For information about this choice, including consent withdrawal, please see our Privacy Policy .

Reviewer #2: **Yes: ** Zalikha Al-Marzouqi

---

## [Editor Report · Acceptance letter]

PONE-D-24-44325R1

PLOS ONE

Dear Dr. Abdi,

I'm pleased to inform you that your manuscript has been deemed suitable for publication in PLOS ONE. Congratulations! Your manuscript is now being handed over to our production team.

Kind regards,

on behalf of

Mr Liknaw Bewket Zeleke

Academic Editor

PLOS ONE